civil engineering/environmental science

underground space reconstruction, engineering repair materials, bacterial induction, nuclear magnetic resonance

**Author for correspondence:**
Hongwei Deng
e-mail: denghw208@126.com

# Experimental study on repair of fractured rock mass by microbial induction technology

Rugao Gao[1,2], Yilin Luo[1,2] and Hongwei Deng[1,2]

[1]College of Resources and Safety Engineering, Central South University, Changsha 410083, People's Republic of China
[2]Hunan Key Laboratory of Mineral Resources Exploitation and Hazard Control for Deep Metal Mines, Changsha 410083, People's Republic of China

 RG, 0000-0002-1840-5298

The surrounding rock mass is often required to have good strength and impermeability in underground engineering. Some grouting methods, such as the chemical grouting method and the cement grouting pressure pump method, are often applied to reform underground environment and improve the engineering performance of rock mass. However, the application of some traditional grouting materials would destroy the original environment in which the project is located. This paper focuses on the repairing effect of *Bacillus pasteurii* composites in fractured rocks. The repairing effect of microbial materials on fractured sandstone is analysed through nuclear magnetic resonance (NMR) and unconfined compression-shearing equipment. The result shows that the longer the repairing time is, the better the effect will be. After 42 days of repairing, the porosity of fractured sandstone decreases by 36.41%, the impermeability increases by 94.62%, and the compressive strength increases by 30.52%. Through the study of reaction mechanisms, this technology has the advantages of mild reaction conditions, no pollution and good environmental compatibility. The application of this technology to the maintenance of geotechnical engineering can provide new ideas for the research and development of new environmental grouting materials and underground space reconstruction technology.

## 1. Introduction

With the development and use of deep resources and space, a number of projects have gradually turned from the surface to underground. How to improve rock mass strength while improving its anti-seepage characteristics has become a hot spot

for the sustainable development of mining and geotechnical engineering. At present, chemical grouting [1], pressure grouting [2], curtain grouting [3] and other means are frequently adopted to improve the strength and anti-seepage characteristics of rock mass. In some studies, some active materials such as volcanic ash, crystals, etc., have also been applied to geotechnical restoration and seepage prevention [4–7]. Apart from these conventional techniques, microbial-induced repair technology has also attracted great attention.

About microbial remediation research, as early as in 1973, Boquet *et al*. [8] found that some microorganisms in rocks have repairing effects on rock cracks, and the induced sediments can still play a long time role after the death of microorganisms. The paper published by Carlos Rodriguez-Navarro *et al*. [9] in 2003 introduced their team's exploration of stone surface repair using yellow bacterium. Stocks *et al*. [10] found that *Bacillus pasteurii* can precipitate calcium carbonate more quickly and cement the sand particles. Terzis *et al*. [11] decomposed soil bacteria by freeze-drying urea. *Sporosarcina pasteurii* introduced the application of microbial-induced carbonate precipitation (MICP). Xiao *et al*. [12] studied the cyclical resistance of calcareous sand and reduced its liquefaction potential, providing a strategy for MICP to improve the cyclical resistance of calcareous sand. Qian *et al*. [13], De Belie & Wang [14] Wang & Qian [15] discussed in detail the research progress of microbial-induced mineralization technology in recent years, and carried out experiments to study the effects of different smearing and injection processes on mineralization results, confirming that the crystal form of calcium carbonate of the deposition is calcite. Moreover, the advantages and disadvantages of active and passive repairing of microorganisms were analysed based on the experiments. Solutions to those disadvantages were put forward accordingly. These studies are aimed at the reaction process of rocks and their repairing effects while that of fractured rock has not been analysed yet, and the repairing mechanism has not been made clear. However, uncertain fractured rock masses often exist where underground resources development and engineering construction projects locate, such as groundwater storage, ionic rare earth mining and restoration, etc. Therefore, the rock mass is required to have good strength and anti-seepage performance. In the view of this, yellow sandstone is applied in this paper to carry out microbial induction tests with prefabricated fractured rock, quantitatively analysing the variation of physical and mechanical properties and permeability parameters before and after the test, and revealing the mechanism of microbial repair of engineering rock mass, so as to provide theoretical basis for its application in practice.

# 2. Experimental materials and schemes

## 2.1. Strain selection and sample preparation

With higher efficiency of precipitating calcium carbonate [16–18], *Bacillus pasteurii* is applied in the experiment. It produces a kind of urease during metabolism, which can decompose urea to form ammonium and carbonate ions. When a certain concentration of calcium ions is contained in the solution, calcium ions will be adsorbed by the cells, thus filling the pore structure. The rock samples are taken from the same intact and unweathered yellow sandstone, whose main compositions are quartz and clay minerals. It was prepared into a standard cylinder with a diameter of $(50 \pm 1)$ mm and a height of $(100 \pm 1)$ mm in line with the *Rules for Rock Experiment of Water Conservancy and Hydropower Engineering* [19]. The water used in the leaching experiment is tap water, which was sealed in plastic buckets prior for further analysis. The pH of the test water is 7.45 and the COD value is 15.04. There are little harmful substances in it.

According to the study by Zhang *et al*. [20,21], when the joint angle in the rock reaches 45°, the uniaxial compressive strength of the specimen will be the minimum. Then it begins to increase in reverse. In the case of confining pressure, it has the greatest impact on the 45° crack. In order to study the effect of MICP on improving the strength and impermeability of fractured rock, a penetrating fissure with a length of 45 mm, a width of 2 mm and a dip angle of 45° is prefabricated on all cylindrical rock samples, as shown in figure 1. Quartz sand mixed with bacteria and nutrient solution was filled into the prefabricated cracks for the repairing experiment.

## 2.2. Experimental schemes

In order to analyse the repairing effect of *Bacillus pasteurii* on the cracked yellow sandstone at different times, the wave velocity of the rock samples was tested by HS-YS4A, a tester of rock acoustic wave

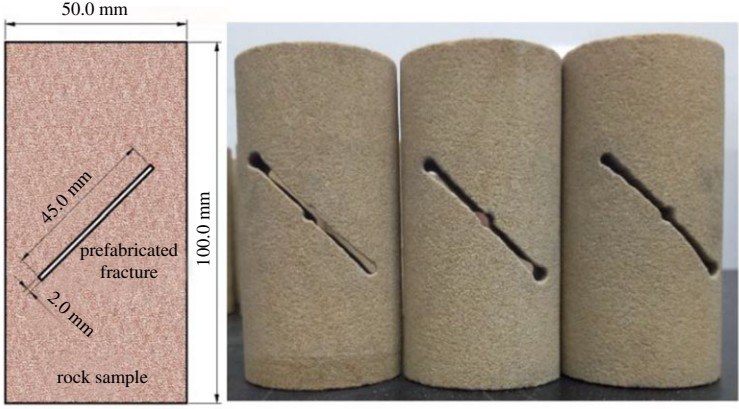

**Figure 1.** Rock samples and the size of prefabricated fissures.

**Table 1.** Experimental grouping.

| no. of group | cultivating time (d) | no. of rock samples | average wave velocity ($\times 10^3$ m s$^{-1}$) | standard deviation |
|---|---|---|---|---|
| A | 0 | A-1, A-2, A-3 | 1.15 | 0.015 |
| B | 7 | B-1, B-2, B-3 | | |
| C | 14 | C-1, C-2, C-3 | | |
| D | 21 | D-1, D-2, D-3 | | |
| E | 28 | E-1, E-2, E-3 | | |
| F | 35 | F-1, F-2, F-3 | | |
| G | 42 | G-1, G-2, G-3 | | |

parameters. Subsequently, 21 rock samples with a similar wave velocity were selected to ensure the same initial state. They were divided into seven groups and tested at different repair periods. The experimental grouping and results of the acoustic parameters are shown in table 1.

Before the experiment, the quartz sand was first placed in the prefabricated crack of the rock sample, then sequentially dried (baking temperature 105°C, drying time 48 h), vacuum saturated (vacuum pressure value 0.1 MPa, pumping time 6 h) and tested by nuclear magnetic resonance (NMR) to obtain the initial porosity and permeability coefficient of the rock samples. As the control group, group A was not processed while the quartz sand in the other samples was taken out of from the fissures. After mixing them with the bacterial liquid and nutrient solution containing urea and $Ca(NO_3)_2$ (both were 2 mol l$^{-1}$ concentration), they were refilled into the prefabricated cracks and a layer of bacterial liquid was quickly brushed on the surface of the rock samples. After that, the rock samples were cultivated at room temperature, and nutrient solution was injected into the cracks from both sides every 2 days until the mixture was saturated, then a layer of bacterial liquid was pasted on the surface of the rock samples again. After each group of samples were repaired, sterilization treatment was first carried out, and NMR and unconfined compression tests followed to obtain the permeability, strength and other parameters of the repaired samples.

# 3. Experimental results and analysis

## 3.1. Repairing effect of microbial-induced carbonate precipitation on the surface of cracked rock samples

In order to observe the repairing effect of MICP on the surface of cracked rock mass, the final repair conditions of the surface of the samples in groups B, C, D and E were recorded, as shown in figure 2, manifesting that the pre-formed fissures of the rock samples are filled with a mixture of quartz sand, bacterial liquid and nutrient solution. Plenty of visible white powder adheres to the crack, the amount

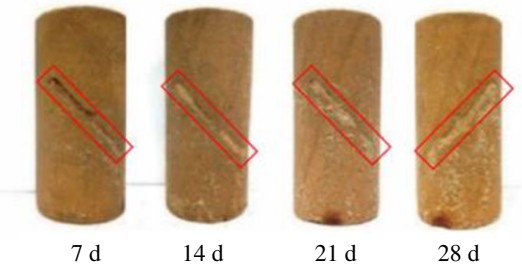

**Figure 2.** Surface repair effect of rock samples at different times.

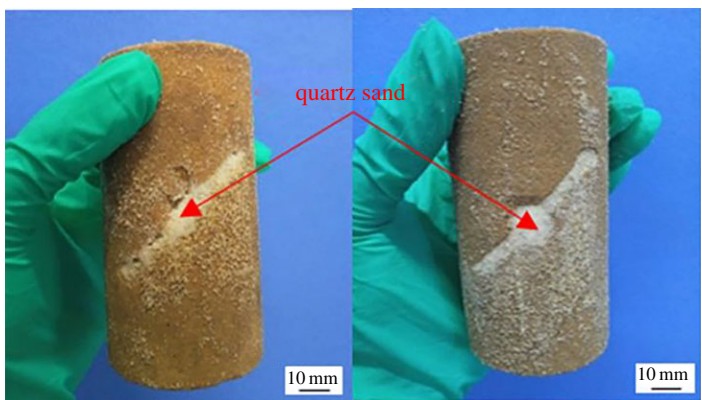

**Figure 3.** Surface repair effect of rock samples in groups F and G.

and coverage area of which significantly increases with time. The quartz sand in the crack is no longer loose, but cemented as a whole by the test products. Prefabricated fissures in the rock samples are gradually blocked. White powder can also be observed to overflow from the crack in groups F and G (figure 3). From what has been mentioned above, a conclusion can be drawn that the longer the MICP test time is, the better the repairing effect on the surface of the cracked yellow sandstone will be.

## 3.2. Effect of microbial-induced carbonate precipitation on permeability of rock samples

The NMR technique is based on the interaction between a magnetic field of hydrogen nuclei and an external magnetic field. Under the influence of a static magnetic field and alternating magnetic field, the H proton in water-saturated specimens will release energy, which is called the NMR signal. The transverse relaxation time T2 distribution can be obtained through the difference of the signals, which directly reflects the variation features of rock pore structure. Distribution of pore fluids is controlled by pore structure of the rock samples. The presence of different pore fluids can be measured by the NMR system (figure 4), which is the basis for evaluating the pore structure by NMR T2 spectrum. Yet, changes of the pore structure will lead to changes in permeability of the rock samples. Estimation of permeability of the rock samples by the NMR system is based on the assumption that permeability improves with the increase of porosity and pore size, based on which, two types of models have been developed: the Coates model and the average T2 model. When all other factors in these models remain constant, the permeability increases as the connected porosity increases. As the MICP test proceeds, the product will cement in the pores. As a result, the pore size inside the samples will change, and the permeability of the rocks will be affected accordingly. According to the research of Jinsoo Uh *et al.* [22], the Coates model is more flexible than the average T2 model. Besides, the permeability measured by the Coates model under the same experimental conditions is more consistent with the actual situation of sandstone. Therefore, the Coates-cutoff model is adopted in this paper to test the permeability of the rock samples. The changes of the seepage performance before and after the repair of the cracked rock samples were analysed according to the corresponding evaluation criteria. The evaluation standard and testing results before and after the repair of the rock samples are shown in tables 2 and 3, respectively.

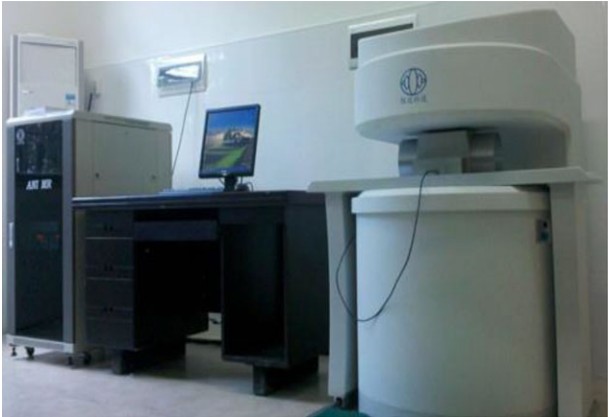

**Figure 4.** NMR equipment.

**Table 2.** Reservoir permeability evaluation standard table.

| grade | millidarcy (mD) | penetrating quality |
|---|---|---|
| 1 | >1000 | excellent |
| 2 | 100–1000 | preferable |
| 3 | 10–100 | general |
| 4 | 1–10 | poor |
| 5 | <1 | worst |

Figure 5 is a trend graph showing the percentage decrease in permeability at different times. It can be seen that the permeability of rock samples declines in varying degrees with the repairing time. The longer it is, the smaller the permeability will be. The percentage decrease in average permeability also increases with time. In the light of the evaluation of the permeability before and after the test, the permeability of the rock samples generally reduces by one to two orders of magnitude. In addition, from the third batch of the rock samples, the permeability all dropped below 10 mD, indicating that the cementation of the MICP test products in the pore structure of the rock sample effectively improved the anti-seepage performance of the rock samples, that is, the effect of using the MICP test to improve the seepage prevention performance of the rock mass is remarkable.

## 3.3. Effect of microbial-induced carbonate precipitation on compressive strength of rock samples

In order to study the effect of MICP on improving rock sample strength, the uniaxial compressive strength of the rock samples was measured by a SHT4206 servo universal testing machine (figure 6) after the nuclear magnetic test. The SHT4206 servo universal testing machine is mainly used for tensile, compression, bending and shearing experiments of rock materials. The maximum experimental force of the performance parameters is 2000 kN, the relative error of the experimental force is ±0.5%, and the displacement measurement resolution is 0.013 mm. The total power is 4 kW.

As can be seen from figure 7, the uniaxial compressive strength of the rock samples at different repairing times is improved to some extent compared to that before the MICP test. The longer the repairing time is, the greater the uniaxial compressive strength will be. The maximum average uniaxial compressive strength increases from 3.08 to 4.02 MPa with an increase rate up to 30.52%. From the variation curves of the average compressive strength, it can be seen that the overall increasing rate follows 'slow-fast-smooth-fast'. The reason why the increase of rock sample strength is slow is that the bacteria are still in a lag phase in the beginning, thus the amount of induced calcium carbonate crystals is too small to form effective cementation. After a period of incubation, the amount of induced calcium carbonate crystals grows rapidly. Therefore, the increasing rate of rock sample strength is improved accordingly. However, because *Bacillus pasteurii* is an aerobic bacterium, the induced calcium carbonic

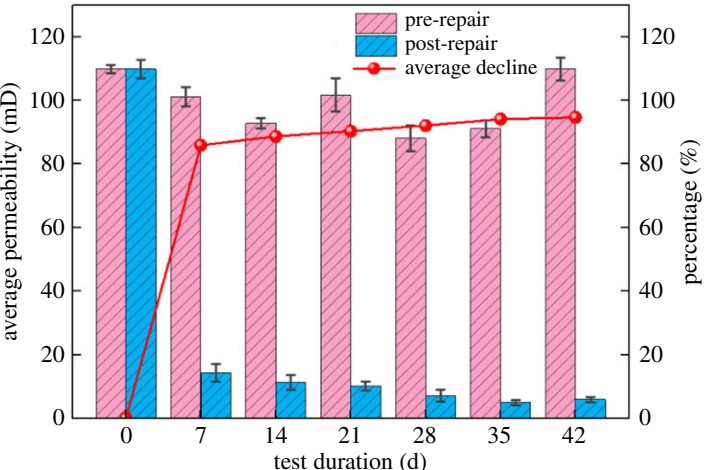

**Figure 5.** Statistics of average permeability and its decline percentage of rock samples.

**Table 3.** Permeability variation statistics of fractured rock samples.

| no. of rock samples | processing time (d) | pre-repair permeability (mD) | post-repair permeability (mD) | average decline percentage (%) | pre-repair penetrating quality | post-repair penetrating quality |
|---|---|---|---|---|---|---|
| A-1 | 0 | 84.58 | | | general | |
| A-2 | | 112.00 | | | preferable | |
| A-3 | | 133.07 | | | preferable | |
| B-1 | 7 | 88.53 | 16.17 | 85.86 | general | general |
| B-2 | | 128.31 | 16.693 | | preferable | general |
| B-3 | | 86.54 | 9.999 | | general | poor |
| C-1 | 14 | 85.43 | 9.512 | 88.65 | general | poor |
| C-2 | | 91.56 | 11.142 | | general | general |
| C-3 | | 101.23 | 13.206 | | preferable | general |
| D-1 | 21 | 101.20 | 10.883 | 90.28 | preferable | general |
| D-2 | | 122.94 | 11.452 | | preferable | general |
| D-3 | | 80.86 | 8.223 | | general | poor |
| E-1 | 28 | 85.24 | 7.745 | 92.07 | general | poor |
| E-2 | | 85.66 | 5.086 | | general | poor |
| E-3 | | 93.26 | 8.667 | | general | poor |
| F-1 | 35 | 80.97 | 7.106 | 94.09 | general | poor |
| F-2 | | 107.90 | 4.727 | | preferable | poor |
| F-3 | | 84.29 | 3.076 | | general | poor |
| G-1 | 42 | 116.384 | 5.323 | 94.62 | preferable | poor |
| G-2 | | 104.93 | 4.351 | | preferable | poor |
| G-3 | | 108.35 | 8.043 | | preferable | poor |

deposits in the gap between the sand grains with the repairing time going by, makes it more difficult for oxygen and nutrient solutions to get into the crack, thus affecting the repairing effect. In the view of this, starting from the fourth batch of rock samples, the way of brushing nutrient solution at the crack end to make it penetrate slowly is abandoned. Instead, micro-syringes are adopted to penetrate into the crack

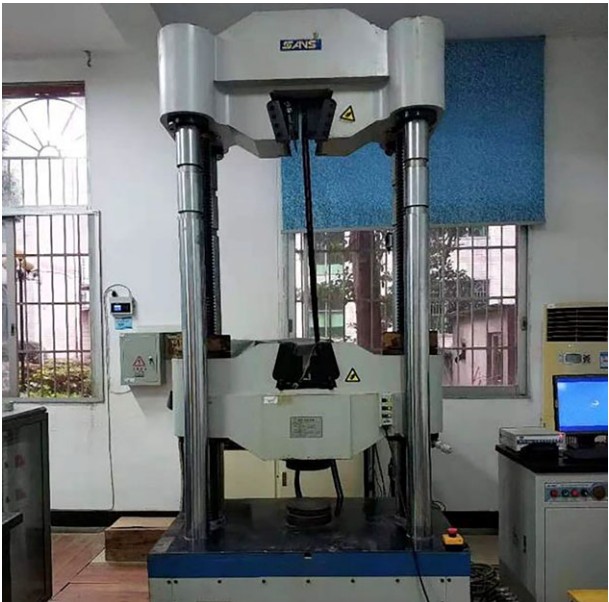

**Figure 6.** SHT4206 servo universal experimental machine.

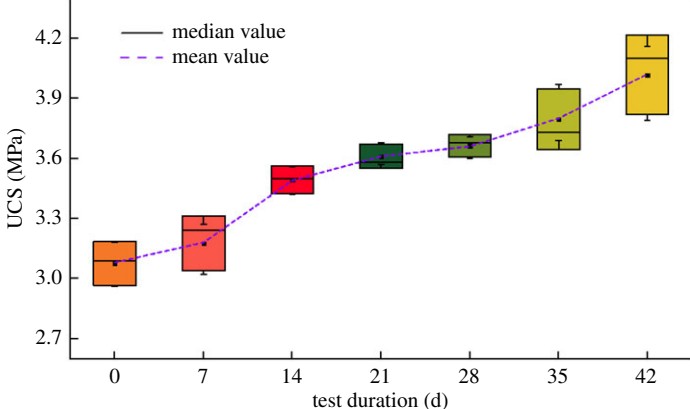

**Figure 7.** Statistics of uniaxial compressive strength of rock samples.

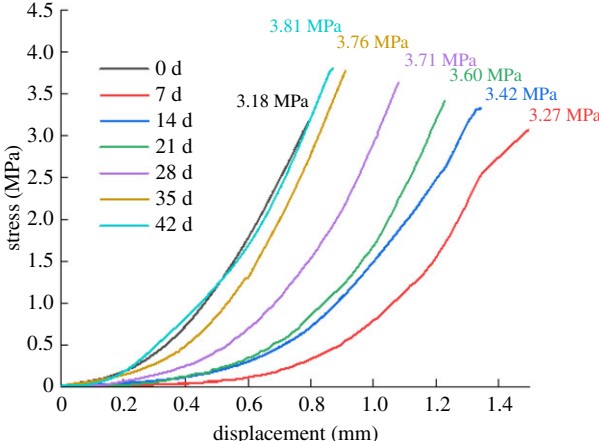

**Figure 8.** Stress and strain curves of rock samples in different periods.

interior to reduce the dropping speed and provide nutritional support for the physiological activities of bacteria. The figure shows that the growth rate of the strength of the fourth and fifth batch of samples is sharply increased after changing the liquid injection mode.

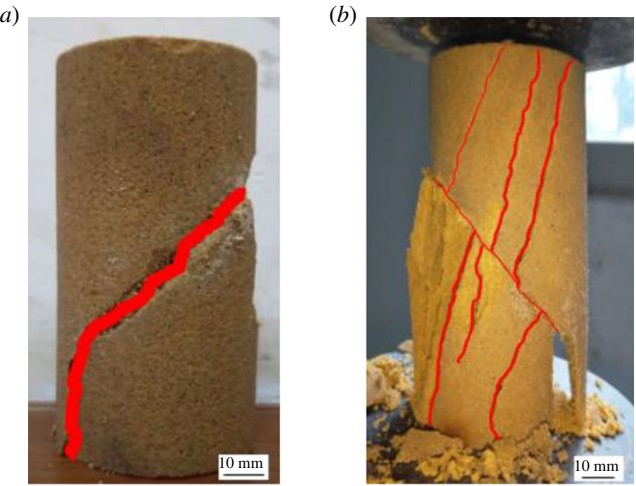

**Figure 9.** Post-peak pressure diagram of rock samples. (*a*) Unsolidified rock sample and (*b*) solidified rock sample.

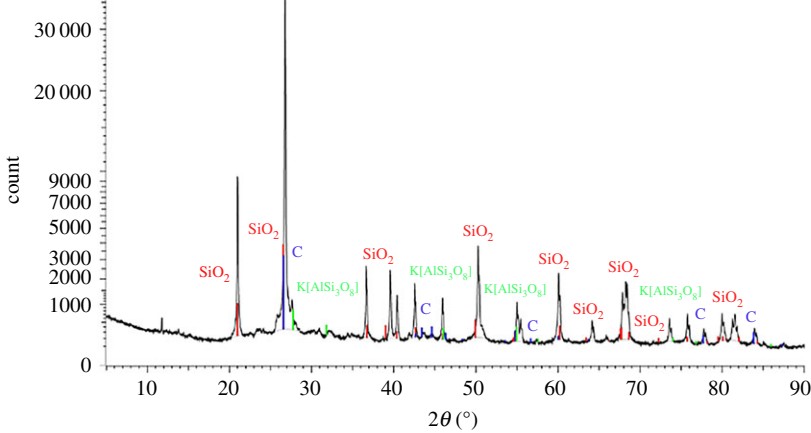

**Figure 10.** Composition analysis before experimental processing.

While improving the strength of rock samples, MICP also changes the stress–strain process and failure mode of rock samples. The stress–strain curves and typical failure modes of the rock at different repairing times are shown in figures 8 and 9, respectively. The stress–strain curves in figure 8 are measured results of A1–G1 group specimens. It can be concluded that, compared to the unrepaired rock, the displacement interval of the micro-crack compaction stage of the rock peak shifted back. The stiffness of the initial specimen is essentially lower than that before repairing, owing to the fact that the bacteria and nutrient solution need to be regularly applied to the cracks and the surface of the rock samples during the test. As time goes by, the effect of bacteria on the fractured yellow sandstone gets better and better, so that the stiffness of the rock samples is increasing, closing to or even exceeding the stiffness of the unsolidified (i.e. not corroded by water) specimens. Figure 9 demonstrates that the unsolidified rock sample scatters immediately after reaching the peak strength, while the solidified rock sample can maintain the basic shape of the rock sample to bear the pressure after being destroyed along the direction of the prefabricated crack. The failure of the rock also transfers from the direction of the prefabricated fracture to the stretching direction.

# 4. Mechanism of microbial-induced carbonate precipitation repairing fractured rock

According to the test results above, MICP significantly improved the strength and anti-seepage performance of the rock samples. In order to reveal the repair mechanism of MICP, an NMR test was carried out on the fractured yellow sandstone. The compositional changes of cemented sand in prefabricated cracks were

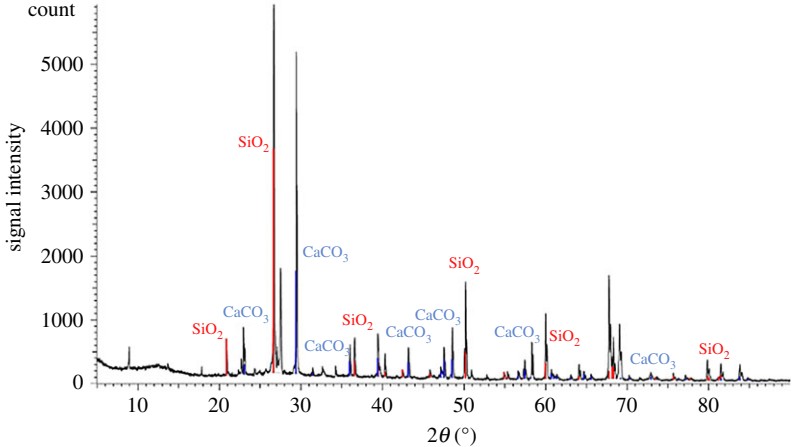

**Figure 11.** Composition analysis after experimental processing.

analysed by X-ray diffraction (XRD) and scanning electron microscope (SEM) to observe the reaction process and mechanism of MICP from the microscopic and mesoscopic points of view.

From the comparative analysis of the composition of the cemented sand before and after the test, the XRD spectra of the sand mixture processed by the MICP test can detect obvious peaks of calcium carbonate, which indicates that a considerable amount of calcium carbonate is produced during the test (figures 10 and 11). The calcium carbonate formed by MICP has a high degree of bonding with the geotechnical material, and the depth of the bonding layer can reach the micrometre level. Calcium carbonate is not only a filler between the pores of the soil, but also a good cementing object. The soil particles are bonded together and a layer of calcium carbonate protective film is deposited on the surface of the rock to improve the durability of these materials. The reinforcement effect is achieved by reducing the void ratio and permeability coefficient of the soil and increasing the strength.

## 4.1. Reaction mechanism

After the compressive strength test of the samples, the cements were removed from them, which were subsequently sampled, metal sprayed and vacuumed before being tested by a NovaNanoSEM230, obtaining SEM images of the cements at different magnifications, as shown in figure 12. It can be seen that the calcium carbonate crystals induced by *Bacillus pasteurii* are attached to the surface of the sand, because *Bacillus pasteurii* produces a kind of active urease during growth [23,24], which can substantially accelerate urea hydrolysis into $NH_3$ and $CO_2$ in the environment. The former will continuously increase the pH value of the surrounding environment so that $CO_2$ is in the presence of $CO_3^{2-}$ which will react with $Ca^{2+}$ in the solution, forming cemented calcium carbonate crystals. The reaction equations are followed as (4.1)–(4.3):

$$Ca^{2+} + Cell \rightarrow Cell \cdot Ca^{2+}, \tag{4.1}$$

$$HCO_3^- + NH_3 \rightarrow NH_4^+ + CO_3^{2-} \tag{4.2}$$

and

$$Cell \cdot Ca^{2+} + CO_3^{2-} \rightarrow Cell \cdot CaCO_3 \downarrow . \tag{4.3}$$

The specific reaction mechanism is shown in figure 12, where $b$ and $c$ are magnified images of the red rectangular area in $a$ and $b$, respectively.

After the formation of calcium carbonate crystals, *Bacillus pasteurii* acts as a nucleus [25,26], adsorbing the calcium carbonate crystals and free solid particles in the environment, so that the calcium carbonate crystals continue to grow and expand. These calcium carbonate crystals either compactly array on the surface of the sand, increasing its particle size or deposit in the gap between sands, forming matrix-embedded skeleton particles, between which a large number of flaky cosmids grow and deposit in highly aggregating condition, forming smooth colloidal connecting keys which greatly strengthen the cementation between sands and between sands and rock. The sand particles are cemented together, presenting good strength properties. When the fractured yellow sandstone is pressed, the cemented

(a)

rock

microbial fillings

rock

(b)

rock

sand particle

(c)

sand particle

$CaCO_3$

Microbial cells: *Bacillus pasteurii*
$NH_2{-}CO{-}NH_2 + H_2O \rightarrow 2HN_3 + CO_2$

$CaCO_3$

$H_2O$

$Ca^{2+}$

$Ca^{2+}$

$H_2O$

$CaCO_3$

$H_2O$

$Ca^{2+}$

$Ca^{2+}$

$H_2O$

$Ca^{2+}$

$CaCO_3$

$NH_3 + H_2O \rightarrow NH_4^+ + OH^-$
$CO_2 + OH^- \rightarrow HCO_3^-$
$Ca^{2+} + HCO_3^- + OH^- \rightarrow CaCO_3 + H_2O$

$CaCO_3$

$Ca^{2+}$

$H_2O$

$H_2O$

$H_2O$

$Ca^{2+}$

$Ca^{2+}$

**Figure 12.** Schematic diagram of reaction mechanism.

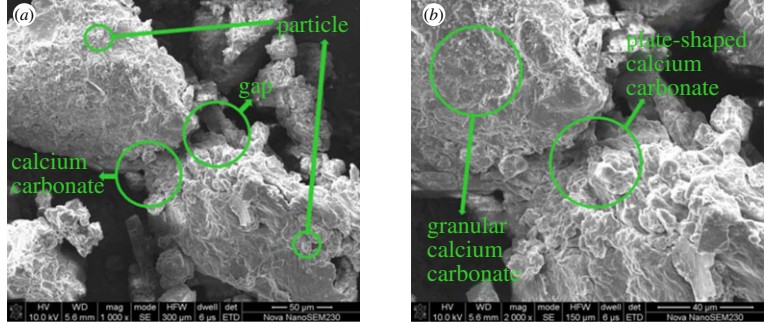

**Figure 13.** SEM of sand mixture. (a) 1000 times and (b) 2000 times.

sand can bear part of the pressure transmitted from the pore wall, thereby improving the strength of the fractured yellow sandstone on the whole. The microstructure changes are shown in figure 13.

## 4.2. Effect of microbial-induced carbonate precipitation on the pores of rock samples

After the calcium carbonate induced by *Bacillus pasteurii* deposits in the pores of the rock, it will cement the particles in the surrounding environment, thus occupying the volume of the original pores, which greatly influences porosity of the rock. The rock samples processed by the MICP test were taken out

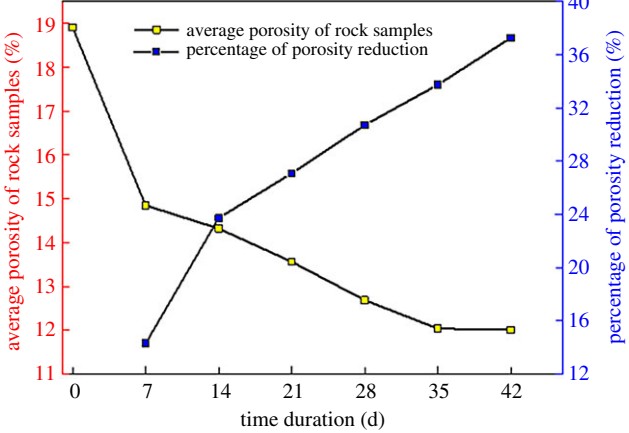

**Figure 14.** Porosity variation statistics of rock samples.

**Table 4.** Porosity variation statistics of rock samples.

| no. of rock samples | processing time (d) | pre-repair porosity (%) | post-repair porosity (%) | percentage decrease in average (%) |
|---|---|---|---|---|
| A-1 | unprocessed | 18.374 | | |
| A-2 | | 18.778 | | |
| A-3 | | 19.590 | | |
| B-1 | 7 | 17.697 | 14.980 | 14.31 |
| B-2 | | 17.107 | 14.729 | |
| B-3 | | 17.194 | 14.842 | |
| C-1 | 14 | 17.736 | 13.590 | 23.73 |
| C-2 | | 18.869 | 14.276 | |
| C-3 | | 19.747 | 15.108 | |
| D-1 | 21 | 18.407 | 14.007 | 27.09 |
| D-2 | | 18.889 | 13.424 | |
| D-3 | | 18.530 | 13.258 | |
| E-1 | 28 | 17.891 | 12.385 | 30.72 |
| E-2 | | 18.513 | 12.448 | |
| E-3 | | 18.532 | 13.226 | |
| F-1 | 35 | 18.634 | 13.397 | 33.74 |
| F-2 | | 18.599 | 11.732 | |
| F-3 | | 17.225 | 10.989 | |
| G-1 | 42 | 19.467 | 12.826 | 36.41 |
| G-2 | | 19.202 | 11.427 | |
| G-3 | | 18.761 | 12.265 | |

of high-temperature sterilization. Afterwards, they were dried and weighed before being saturated. After that, the porosity was measured by an NMR system. Table 4 and figure 14 show the porosity statistics of the fractured yellow sandstone at different repair times.

The figure shows that after the MICP repairing, the porosity of each rock sample decreases to some extent, because the bacteria will move in the original pores of the rock, and the induced sediment will fill the original pore space, so that the volume of the pores of the rock sample is greatly reduced. The average porosity of the rock samples decreases with the repairing time. The average porosity of the rock samples can be reduced to 11% in five weeks, with a dropping rate of 36.34%, signifying that the longer

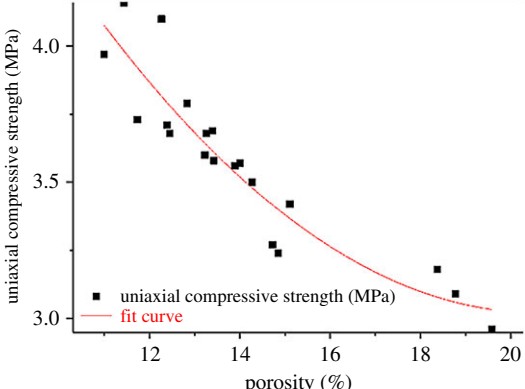

**Figure 15.** Porosity-strength correlation analysis.

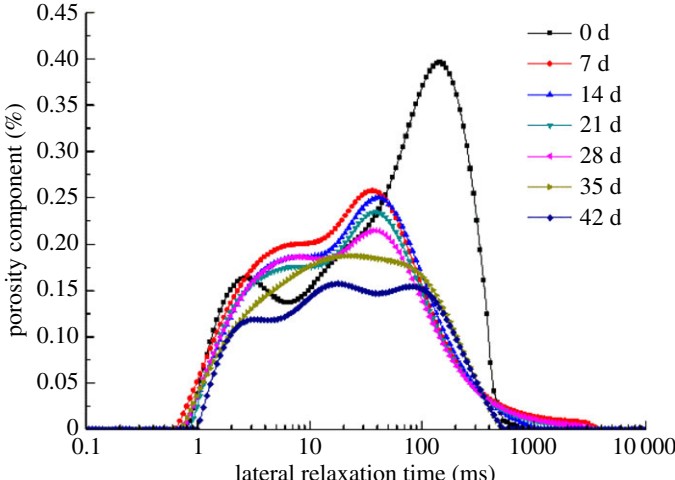

**Figure 16.** T2 spectrum distribution curves of rock samples in different periods.

the time, the more the porosity of the fractured yellow sandstone decreases, which is exactly the same as the change rule of the MICP repairing effect. Furthermore, from the change curves of percentage of porosity reduction of the rock samples, it can be concluded that the repair rate of the bacterial solution declines with the increase of repair time, mainly owing to the fact that *Bacillus pasteurii* is an aerobic bacterium, and the number of bacteria acting near the oxygen-rich pores is large, so even from the fourth batch of the rock samples, the way nutrient solution is provided is altered, the subsequent production of calcium carbonate and its cement will still preferentially accumulate in the pores, unable to enter the depth of the pores, degrading the rate of test products plugging pores of the rock samples.

In the MICP test, the repair effect of calcium carbonate can reach the micrometre level. This effect can be used to improve the strength by filling the cracks and pores inside the rock and soil. NMR can visually show the porosity change of the rock mass, and the actual effect of the MICP can be more accurately expressed through the correlation analysis with the intensity gain. To study the relationship between the decrease of porosity and the increase of uniaxial compressive strength during the process of microbial-induced calcium carbonate deposition, the porosity of 21 fractured yellow sandstones was treated as the independent variable, while its uniaxial compressive strength the dependent variable. The contrast fitting is performed by various functions of the ORIGIN software. It is found that the squared sum of the residuals adopting the polynomial function is only 0.2635, and the value of $R^2$ is 0.862, which means the fitting effect is preferable:

$$y = 7.869 - 0.47x + 0.114x^2, \tag{4.4}$$

where $y$ is the porosity of the fractured yellow sandstone, and $x$ is its uniaxial compressive strength. The specific fitting effect is shown in figure 15.

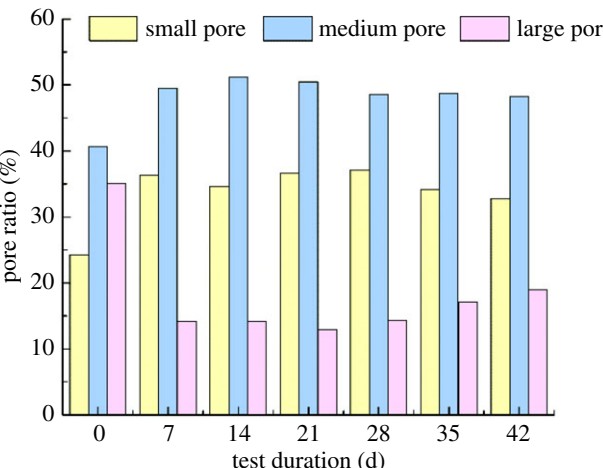

**Figure 17.** Variation of pore size ratio of rock samples in different periods.

The above figure suggests that the uniaxial compressive strength of the rock samples decreases with the increase of porosity, providing mutual verification with the experiment that the uniaxial compressive strength of the fractured yellow sandstone increases with the decrease of porosity. It also proves the feasibility of reducing the porosity of the rock samples to improve its strength and the importance of studying the variation law of pore structure inside the rock samples.

The NMR T2 spectrum distribution curves of group G at different repairing times were measured so as to study the change of pore structure inside the rock samples during microbial-induced repair, as shown in figure 16. The NMR T2 spectrum distribution curves reflect the proportion of different pores in the samples, wherein the smaller the value of T2 is, the smaller the pore will be. The figure implies that as the repairing time increases, the highest peak value of T2 becomes smaller. What is more, the transverse relaxing time corresponding to the peak shifts to the left, and the difference between the peak and trough narrowed crucially, suggesting that the size and number of large pores in the rock samples both reduced while homogeneity of the rock samples is improved. In order to more accurately analyse the variation law of pore distribution inside the rock samples after the test, pores with a transverse relaxation time of less than 10 ms are considered as small ones, those between 10 and 100 ms are medium ones, and over 100 ms are large ones by referring to the practices of Lin [27] and Zhou [28]. Combined with the T2 spectrum distribution curves in figure 16, the spectral areas corresponding to each type of pores are calculated, the proportions of which are as shown in figure 17. It can be seen that the unprocessed rock samples have the highest proportion of medium pores, followed by the large pores and the smallest proportion of small pores. The proportion of large pores in the rock samples decreases sharply after the processing, suggesting that with the cementation of the test products, the original space of the pores is blocked, and pore diameters inside the rock samples are reduced. Some of the large pores are converted into medium and small pores as well, dwindling the number of large pores while multiplying that of small and medium pores.

Because the MICP technology can greatly reduce diameters of the pores inside the rock samples, the pore throat radius is diminished. Hence, the seepage performance of the fractured yellow sandstone is essentially improved. Additionally, the MICP technology changes the pore structure of the rock samples, so that its homogeneity is improved and internal defects are reduced, and the bearing capacity of the fractured rock itself is greatly enhanced.

## 4.3. Fractal geometric expression and fractal dimension of nuclear magnetic resonance T2 spectrum

According to the fractal geometry principle, if the internal pore size distribution of the rock samples conforms to the fractal structure, the number of pores whose diameter is larger than $r$ is assumed as $N\,(>r)$, then the relationship between $N$ and $r$ follows the power function expression [29]:

$$N(> r) = \int_{r}^{r_{\max}} P(r)\, \mathrm{d}r = a r^{-D},\tag{4.5}$$

where $r_{\max}$ is the radius of the maximum pore inside the rock sample; $P(r)$ is the density of pore diameter distribution; $a$ is the proportional constant and $D$ is the pore fractal dimension.

**Table 5.** Fractal dimension statistics of rock samples in different periods.

| cultivating time (d) | K | D | $R^2$ |
| --- | --- | --- | --- |
| 0 | 0.426 | 2.574 | 0.530 |
| 7 | 0.415 | 2.585 | 0.668 |
| 14 | 0.403 | 2.597 | 0.586 |
| 21 | 0.360 | 2.640 | 0.599 |
| 28 | 0.366 | 2.634 | 0.594 |
| 35 | 0.377 | 2.623 | 0.584 |
| 42 | 0.399 | 2.601 | 0.597 |

Derivating $r$ in equation (4.5), the expression of the density of pore diameter distribution is obtained as

$$P(r) = \frac{\mathrm{d}N(>r)}{\mathrm{d}r} = a'r^{D-1}, \tag{4.6}$$

where $a' = -Da$ is a proportional constant.

The cumulative volume of pores whose diameter is smaller than $r$ can be expressed as

$$V(<r) = \int_{r_{\min}}^{r} P(r)ar^3\mathrm{d}r, \tag{4.7}$$

where $a$ is a constant related to the shape of the pore (i.e. when the pore is a cube, $a = 1$; when the pore is a sphere, $a = 4\pi/3$; $r_{\min}$ is the minimum pore radius).

Substituting equation (4.6) into equation (4.7), the following equation can be obtained:

$$V(<r) = a''(r^{3-D} - r^{3-D_{\min}}). \tag{4.8}$$

Hence, the total pore volume inside the rock sample is

$$Vs = V(<r_{\min}) = a''(r^{3-D_{\max}} - r^{3-D_{\min}}). \tag{4.9}$$

Thus, the expression for the cumulative pore volume fraction whose pore diameter is smaller than $r$ is obtained as follows:

$$Sv = \frac{V(<r)}{Vs} = \frac{r^{3-D} - r_{\min}^{3-D}}{r_{\max}^{3-D} - r^{3-D}}. \tag{4.10}$$

Because $r_{\min}$ is much smaller than $r_{\max}$, the above formula can be simplified as

$$Sv = \frac{r^{3-D}}{r_{\max}^{3-D}}. \tag{4.11}$$

Equation (4.10) is a fractal geometric formula for the pore diameter distribution.

In the NMR experiment, the relaxation time $T_2$ increases linearly with the increase of pore diameter. Therefore, $T_2$ implies the pore radius. Consequently, equation (4.11) can be transformed into

$$Sv = \left(\frac{T_{2\,\max}}{T_2}\right)^{D-3}, \tag{4.12}$$

where $Sv$ is the cumulative volume of pores with a transverse relaxation time of less than the percentage of the total pore volume. Thus, an approximate fractal geometry formula for the NMR spectrum is obtained.

Equation (4.13) is obtained after taking the logarithm of the two sides of equation (4.12):

$$\lg Sv = (3 - D)\lg T_2 + (D - 3)\lg T_{2\max}. \tag{4.13}$$

This formula shows that the internal pore structure of the rock samples has obvious fractal features, and the linear correlation between the NMR relaxation table can be verified by graphic or regression

analysis. The magnetic fractal dimension of the pore core is calculated according to the coefficient of the regression equation. The maximum relaxation time and the correlation coefficient given by the regression analysis can explain the reliability of the pore fractal structure. The fractal dimension and correlation coefficient of rock samples in different periods is shown in table 5.

The table indicates that the fractal dimension of the rock sample first increases and then decreases, which means that as the test time increases, the pore structure inside the rock sample first becomes complicated and then simple. This is because the prefabricated crack inside the rock sample is a single through crack. As the sand sample mixture is filled into the crack, the original single through crack is isolated into a myriad of new small pores, along with the induced calcium carbonate. It is also continuously cemented and deposited in the fissures. The original small pores in the rock samples are filled, the pore size of the medium and large pores is gradually reduced, the homogeneity of the rock samples is improved, and the pore structure is gradually simplified.

# 5. Conclusion

Microbial-induced calcium carbonate deposition repairing technology is to decompose urea into carbonate ions and ammonium ions by urease produced by microbial metabolism. It merges with calcium ions at the repairing spot, generating calcium carbonate which deposits at the defect site to repair the crack. The method is environmentally friendly and of great significance for the sustainable development of underground geotechnical engineering. In this study, the effect of *Bacillus pasteurii* on the repairing of yellow sandstone fissures was studied by combining a macro-mechanical test and permeability analysis, as well as the analysis of micro-structure characteristics and repair mechanism. Based on the microscopic pore parameters, the correlation of microbial repairing effects was established. The major conclusions are as follows.

(i) The research shows that the MICP technology works effectively on the repairing of fractured yellow sandstone. The seepage performance of the rock specimen improved by 94.62% in 42 days, and the compressive strength elevated from 3.07 to 4.02 MPa with an increase rate up to 30.52%. The cementation effect of the induced calcium carbonate in the pore structure of the rock can effectively improve its anti-seepage performance and greatly promote the strength of the fractured yellow sandstone. Moreover, it can repair the integrity of the rock to a certain extent and improve its post-peak pressure bearing capacity.

(ii) Combined with XRD and SEM, the formation and mechanism of fillers in the microbial remediation process were analysed. The results demonstrate that the filling of fissures by fractured calcium carbonate can significantly affect the mechanical properties and permeability of rocks. Based on the measurement of T2 spectral distribution by NMR, the correlation between porosity and compressive strength was established. After 42 days of repair, the porosity of the fractured yellow sandstone decreased by 36.41%. As the porosity of the rock sample decreased, the uniaxial compressive strength of the rock sample increased. Combined with the fractal characteristics analysis of the pore structure inside the fractured rock, it can be concluded that the good cementation of *Bacillus pasteurii* produces a good bearing capacity of the sand while the product is deposited in the pores, reducing the pore throat radius of the fractured yellow sandstone, and porosity changes the structure of the seepage channel inside the rock and improves the seepage prevention performance of the cracked rock.

(iii) Based on macroscopic and microscopic microbial-induced calcium carbonate deposition mechanisms, this paper studied the repair effect of MICP technology on fractured yellow sandstone by introducing rock mechanics and seepage tests. Combined with the microscopic pore distribution and fractal characteristics, the experimental data were used to analyse the quantitative characterization of microbial remediation effects in different periods, providing useful insights for the sustainable development of mining and geotechnical engineering. Owing to the complexity of the rock mass structure and water environment in the project, extensive research still needs to be done on the application of microbial induction technology in the repair of engineering fractured rock mass.

Ethics. We have obtained publication permission from all the authors, and we declare that the present experiments and manuscript were performed in accordance with the standard of academic conduct from Chinese academic societies. No experiments in this study included human studies or field studies on animals.

Data accessibility. The datasets supporting this article have been uploaded as part of the electronic supplementary material.

Authors' contributions. R.G.G. and Y.L.L. designed the experimental process. Y.L.L. performed the experimental work. R.G.G. tested the strength of all samples and finished other tests. R.G.G. and H.W.D collected and analysed the data. R.G.G. and H.W.D. interpreted the results and wrote the manuscript. All authors gave final approval for publication.

Competing interests. We declare that we have no competing interests.

Funding. This research was supported by the National Natural Science Fund (grant no. 51874352) funded by the Ministry of Science and Technology of the People's Republic of China.

Acknowledgements. We thank the instructional support specialist Modern Analysis and Testing Central of Central South University. Finally, we thank the anonymous reviewers for their helpful comments.

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
