## [Reviewer comments · Royal Society Open Science]

Review History

RSOS-191318.R0 (Original submission)

Review form: Reviewer 1

Is the manuscript scientifically sound in its present form?

Yes

Are the interpretations and conclusions justified by the results?

Yes

Is the language acceptable?

Yes

Do you have any ethical concerns with this paper?

No

Have you any concerns about statistical analyses in this paper?

No

Recommendation?

Accept with minor revision (please list in comments)

Comments to the Author(s)

Article deals with an important issue of repairing fractured rocks by using *Bacillus pasteurii* composites. The repairing effect of microbial materials on fractured sandstone is analysed by means of nuclear magnetic resonance (NMR) and unconfined compression-shearing equipment. Authors were successful in using the analytical and testing methods. However, some comments need to be rectified:

1. Introduction part needs to be extended to include other used materials for repairing stones, bricks and concrete. The following research have already discussed this topic:(a) Al-Kheetan, M.J., Rahman, M.M. and Chamberlain, D.A., 2018. A novel approach of introducing crystalline protection material and curing agent in fresh concrete for enhancing hydrophobicity. *Construction and Building Materials*, 160, pp.644-652. (b) Al-Kheetan, M.J., Rahman, M.M. and Chamberlain, D.A., 2018. Remediation and protection of masonry structures with crystallising moisture blocking treatment. *International Journal of Building Pathology and Adaptation*, 36(1), pp.77-92. (c) Al-Kheetan, M.J., Rahman, M.M. and Chamberlain, D.A., 2019. Moisture evaluation of concrete pavement treated with hydrophobic surface impregnants. *International Journal of Pavement Engineering*, pp.1-9. (d) Al-Kheetan, M.J. and Rahman, M.M., 2019. Integration of Anhydrous Sodium Acetate (ASAc) into Concrete Pavement for Protection against Harmful Impact of Deicing Salt. *JOM*, pp.1-11. (e) Al-Kheetan, M.J., Rahman, M.M. and Chamberlain, D.A., 2019. Fundamental interaction of hydrophobic materials in concrete with different moisture contents in saline environment. *Construction and Building Materials*, 207, pp.122-135. (f) Al-Kheetan, M.J., Rahman, M.M. and Chamberlain, D.A., 2017. Influence of crystalline admixture on fresh concrete to develop hydrophobicity, *Transportation Research Board 96th Annual Meeting*, Washington D.C. (No. 17-02487).
2. Please include error bars in Figures 5, 7 and 8.
3. In section 4 (Mechanism of MICP repairing fractured rock), please extend the discussion and explain more on the influence of calcium carbonate and its presence in the cemented sand.
4. Please make the annotation on SEM images clearer. Text in the images are barely read.
5. Manuscript needs proofreading. Some grammar mistakes were detected.

I would like to recommend this article for publication after authors rectify the previous comments.

Review form: Reviewer 2

Is the manuscript scientifically sound in its present form?

Yes

Are the interpretations and conclusions justified by the results?

Yes

Is the language acceptable?

Yes

Do you have any ethical concerns with this paper?

No

Have you any concerns about statistical analyses in this paper?

No

Recommendation?

Accept with minor revision (please list in comments)

Comments to the Author(s)

This research on MICP technology is detailed and meaningful. By combining the analysis of microstructure features with macroscopic performance, the authors present reliable results. Therefore, I suggest that this article be accepted for publication.

1. The contents of Table 1 need to be revised.
2. The author should supplement the parameters of the water used in the test to provide support for the reliability of the test results.
3. Page 6, Fig.4 should be changed to Fig.5
4. The author should explain why the correlation between strength and porosity is analyzed.
5. Porosity studies are meaningful for the repair of microscopic structures. Please supplement the test principle of nuclear magnetic resonance equipment.
6. The conclusion can be appropriately streamlined.

Decision letter (RSOS-191318.R0)

22-Oct-2019

Dear Dr Gao,

On behalf of the Editors, I am pleased to inform you that your Manuscript RSOS-191318 entitled "Experimental Study on Repair of Fractured Rock Mass by Microbial Induction Technology" has been accepted for publication in Royal Society Open Science subject to minor revision in accordance with the referee suggestions. Please find the referees' comments at the end of this email.

The reviewers and handling editors have recommended publication, but also suggest some minor revisions to your manuscript. Therefore, I invite you to respond to the comments and revise your manuscript.

- Ethics statement

- Data accessibility

It is a condition of publication that all supporting data are made available either as supplementary information or preferably in a suitable permanent repository. The data accessibility section should state where the article's supporting data can be accessed. This section should also include details, where possible of where to access other relevant research materials such as statistical tools, protocols, software etc can be accessed. If the data has been deposited in

an external repository this section should list the database, accession number and link to the DOI for all data from the article that has been made publicly available. Data sets that have been deposited in an external repository and have a DOI should also be appropriately cited in the manuscript and included in the reference list.

<http://datadryad.org/submit?journalID=RSOS&manu=RSOS-191318>

- **Competing interests**

- **Authors' contributions**

- **Acknowledgements**

- **Funding statement**

Because the schedule for publication is very tight, it is a condition of publication that you submit the revised version of your manuscript before 31-Oct-2019. Please note that the revision deadline will expire at 00.00am on this date. If you do not think you will be able to meet this date please let me know immediately.

Kind regards,
Lianne Parkhouse
Editorial Coordinator
Royal Society Open Science
openscience@royalsociety.org

on behalf of Dr Maria Charalambides (Associate Editor) and R. Kerry Rowe (Subject Editor)
openscience@royalsociety.org

Reviewer comments to Author:

Reviewer: 1
Comments to the Author(s)

Article deals with an important issue of repairing fractured rocks by using *Bacillus pasteurii* composites. The repairing effect of microbial materials on fractured sandstone is analysed by means of nuclear magnetic resonance (NMR) and unconfined compression-shearing equipment. Authors were successful in using the analytical and testing methods. However, some comments need to be rectified:

1. Introduction part needs to be extended to include other used materials for repairing stones, bricks and concrete. The following research have already discussed this topic:(a) Al-Kheetan, M.J., Rahman, M.M. and Chamberlain, D.A., 2018. A novel approach of introducing crystalline protection material and curing agent in fresh concrete for enhancing hydrophobicity. *Construction and Building Materials*, 160, pp.644-652. (b) Al-Kheetan, M.J., Rahman, M.M. and Chamberlain, D.A., 2018. Remediation and protection of masonry structures with crystallising moisture blocking treatment. *International Journal of Building Pathology and Adaptation*, 36(1), pp.77-92. (c) Al-Kheetan, M.J., Rahman, M.M. and Chamberlain, D.A., 2019. Moisture evaluation of concrete pavement treated with hydrophobic surface impregnants. *International Journal of Pavement Engineering*, pp.1-9. (d) Al-Kheetan, M.J. and Rahman, M.M., 2019. Integration of Anhydrous Sodium Acetate (ASAc) into Concrete Pavement for Protection against Harmful Impact of Deicing Salt. *JOM*, pp.1-11. (e) Al-Kheetan, M.J., Rahman, M.M. and Chamberlain, D.A., 2019. Fundamental interaction of hydrophobic materials in concrete with different moisture contents in saline environment. *Construction and Building Materials*, 207, pp.122-135. (f) Al-Kheetan, M.J., Rahman, M.M. and Chamberlain, D.A., 2017. Influence of crystalline admixture on fresh concrete to develop hydrophobicity, *Transportation Research Board 96th Annual Meeting*, Washington D.C. (No. 17-02487).

2. Please include error bars in Figures 5, 7 and 8.

3. In section 4 (Mechanism of MICP repairing fractured rock), please extend the discussion and explain more on the influence of calcium carbonate and its presence in the cemented sand.

4. Please make the annotation on SEM images clearer. Text in the images are barely read.

5. Manuscript needs proofreading. Some grammar mistakes were detected.

I would like to recommend this article for publication after authors rectify the previous comments.

Reviewer: 2

Comments to the Author(s)

This research on MICP technology is detailed and meaningful. By combining the analysis of microstructure features with macroscopic performance, the authors present reliable results. Therefore, I suggest that this article be accepted for publication.

1. The contents of Table 1 need to be revised.
2. The author should supplement the parameters of the water used in the test to provide support for the reliability of the test results.
3. Page 6, Fig.4 should be changed to Fig.5
4. The author should explain why the correlation between strength and porosity is analyzed.
5. Porosity studies are meaningful for the repair of microscopic structures. Please supplement the test principle of nuclear magnetic resonance equipment.
6. The conclusion can be appropriately streamlined.

Author's Response to Decision Letter for (RSOS-191318.R0)

See Appendix A.

Decision letter (RSOS-191318.R1)

29-Oct-2019

Dear Dr Gao,

I am pleased to inform you that your manuscript entitled "Experimental Study on Repair of Fractured Rock Mass by Microbial Induction Technology" is now accepted for publication in Royal Society Open Science.

Kind regards,

on behalf of Dr Maria Charalambides (Associate Editor) and R. Kerry Rowe (Subject Editor)
openscience@royalsociety.org

Appendix A

Reviewer: 1

Comments to the Author(s)

Article deals with an important issue of repairing fractured rocks by using *Bacillus pasteurii* composites. The repairing effect of microbial materials on fractured sandstone is analysed by means of nuclear magnetic resonance (NMR) and unconfined compression–shearing equipment. Authors were successful in using the analytical and testing methods. However, some comments need to be rectified:

1. Introduction part needs to be extended to include other used materials for repairing stones, bricks and concrete. The following research have already discussed this topic: (a) Al-Kheetan, M. J., Rahman, M. M. and Chamberlain, D. A., 2018. A novel approach of introducing crystalline protection material and curing agent in fresh concrete for enhancing hydrophobicity. *Construction and Building Materials*, 160, pp.644–652. (b) Al-Kheetan, M. J., Rahman, M. M. and Chamberlain, D. A., 2018. Remediation and protection of masonry structures with crystallising moisture blocking treatment. *International Journal of Building Pathology and Adaptation*, 36(1), pp.77–92. (c) Al-Kheetan, M. J., Rahman, M. M. and Chamberlain, D. A., 2019. Moisture evaluation of concrete pavement treated with hydrophobic surface impregnants. *International Journal of Pavement Engineering*, pp.1–9. (d) Al-Kheetan, M. J. and Rahman, M. M., 2019. Integration of Anhydrous Sodium Acetate (ASAc) into Concrete Pavement for Protection against Harmful Impact of Deicing Salt. *JOM*, pp.1–11. (e) Al-Kheetan, M. J., Rahman, M. M. and Chamberlain, D. A., 2019. Fundamental interaction of hydrophobic materials in concrete with different moisture contents in saline environment. *Construction and Building Materials*, 207, pp.122–135. (f) Al-Kheetan, M. J., Rahman, M. M. and Chamberlain, D. A., 2017. Influence of crystalline admixture on fresh concrete to develop hydrophobicity, *Transportation Research Board 96th Annual Meeting*, Washington D.C. (No. 17–02487).

Author's reply: We have modified the introduction and added some of the latest references.

2. Please include error bars in Figures 5, 7 and 8.

Author's reply: We have modified these figures, and we have added error bars in Figure 5 and Figure 7. Figure 8 shows the stress–strain curve of the test piece, which mainly reflects the peak intensity change under different conditions. This is a relatively straightforward result, so the error bars are not added.

Fig.5 Statistics of Average permeability and its decline percentage of rock samples

Fig. 7 Statistics of uniaxial compressive strength of rock samples

Fig.8 Stress and strain curves of rock samples in different periods

3. In section 4 (Mechanism of MICP repairing fractured rock), please extend the discussion and explain more on the influence of calcium carbonate and its presence in the cemented sand.

Author's reply: We have supplemented this section as follows:

Composition analysis before experimental processing

Composition analysis after experimental processing

From the comparative analysis of the composition of the cemented sand before and after the test, the XRD spectra of the sand mixture processed by MICP test can detect obvious peaks of calcium carbonate, which indicates that a considerable amount of calcium carbonate is produced during the test. The calcium carbonate formed by MICP has a high degree of bonding with the geotechnical material, and the depth of the bonding layer can reach the micron level. Calcium carbonate is not only a filler between the pores of the soil, but also a good cementing object. The soil particles are bonded together and a layer of calcium carbonate protective film is deposited on the surface of the rock to improve the durability of these materials. The reinforcement effect is achieved by reducing the void ratio and permeability coefficient of the soil and increasing the strength.

4. Please make the annotation on SEM images clearer. Text in the images are barely read.

Author's reply: We have re-edited the images and text in them.

Fig.13 SEM of sand mixture

5. Manuscript needs proofreading. Some grammar mistakes were detected.

Author's reply: We have revised some of the contents of the manuscript based on the comments.

I would like to recommend this article for publication after authors rectify the previous comments.

Reviewer: 2

Comments to the Author(s)

This research on MICP technology is detailed and meaningful. By combining the analysis of microstructure features with macroscopic performance, the authors present reliable results. Therefore, I suggest that this article be accepted for publication.

1. The contents of Table 1 need to be revised.

Author reply: Modified

Table1 Experimental grouping

No. of Cultivating group	time (d)	No. of rock samples	Average wave velocity ($\times 10^3$ m/s)	Standard deviation
A	0	A-1、A-2、A-3		
B	7	B-1、B-2、B-3		
C	14	C-1、C-2、C-3		
D	21	D-1、D-2、D-3	1.15	0.015
E	28	E-1、E-2、E-3		
F	35	F-1、F-2、F-3		
G	42	G-1、G-2、G-3		

2. The author should supplement the parameters of the water used in the test to provide support for the reliability of the test results.

Author's reply: We have supplemented this section as follows:

The water used in the leaching experiment is tap water, which was sealed in plastic buckets prior for further analysis. The pH of the test water is 7.45 and the COD value is 15.04. There is little harmful substance in it.

3. Page 6, Fig.4 should be changed to Fig.5

Author reply: Modified

4. The author should explain why the correlation between strength and porosity is analyzed.

Author's reply: We have supplemented this section as follows:

In the MICP test, the repair effect of calcium carbonate can reach the micron level. This effect can be used to improve the strength by filling the cracks and pores inside the rock and soil. NMR can visually show the porosity change of the rock mass, and the actual effect of the MICP can be more accurately expressed through the correlation analysis with the intensity gain.

5. Porosity studies are meaningful for the repair of microscopic structures. Please supplement the test principle of nuclear magnetic resonance equipment.

Author's reply: We have supplemented this section as follows:

NMR technique is based on the interaction between magnetic field of hydrogen nuclei and external magnetic field. Under the influence of static magnetic field and alternating magnetic field, the H proton in water-saturated specimens will release energy, which is called NMR signal. The transverse relaxation time T2 distribution can be obtained through the difference of the signals, which directly reflects the variation features of rock pore structure. Distribution of pore fluids is controlled by pore structure of the rock samples. The presence of different pore fluids can be measured by nuclear magnetic resonance system, which is the basis for evaluating the pore structure by NMR T2 spectrum.

6. The conclusion can be appropriately streamlined.

Author's reply: We have made some changes to this section.